## [Transparent Peer Review file · Nature Communications]

Substituent-induced oxidation-reduction molecular organic junction for interfacial hydrogen peroxide photosynthesis

Corresponding Author: Professor Junhao Qin

Version 0:

Reviewer comments:

Reviewer #1

(Remarks to the Author)

The authors constructed a covalent triazine framework (CTF)-based catalysts featuring spatially separated oxidation and reduction centers. These catalysts exhibit exceptional performance in the photocatalytic synthesis of H₂O₂ and facilitate the direct oxidation of As³⁺ to As⁵⁺ using in-situ generated H₂O₂. However, several aspects require further attention:

- (1) Although the introduction outlines the design concept of spatially separated redox centers and identifies the corresponding molecular structures, direct evidence confirming their respective roles as oxidation and reduction sites is lacking. Additional experimental or characterization data are necessary to substantiate this claim.
- (2) The reported H₂O₂ production activity was measured under simulated sunlight using a full-spectrum xenon lamp at 150 mW·cm⁻² without an AM 1.5G filter. To accurately reflect solar conditions, performance should be evaluated under standard AM 1.5G irradiation, and the solar-to-chemical conversion (SCC) efficiency should be calculated to enable comparison with other catalytic systems.
- (3) Given the light absorption of catalysts up to 600 nm, measuring the apparent quantum yield (AQY) at specific wavelengths across the 360-600 nm range is recommended. Comparative AQY values against established high-performance catalysts would strengthen the performance assessment.
- (4) While the catalyst demonstrates high H₂O₂ production activity, direct comparative data with existing systems are not provided. Inclusion of such comparisons would better contextualize its performance and highlight any competitive advantages.
- (5) Although multiple characterization techniques were employed to verify the catalyst structure, solid-state NMR (¹³C, ¹H, and ¹⁹F) spectroscopy could provide further insight into the chemical environment and structural integrity.
- (6) The authors suggest that the catalyst can adsorb O₂ from air (Figure 4a). O₂-temperature-programmed desorption (O₂-TPD) experiments are recommended to quantitatively assess O₂ adsorption capacity and offer more direct evidence.
- (7) Figure 4c indicates a high H₂O₂ production rate over 6 hours, with an approximately linear increase and no sign of plateau. Extending the reaction duration would help determine the maximum H₂O₂ yield, while post-reaction characterization of the catalyst would aid in evaluating its stability.
- (8) Although isotope labeling experiments confirm the H₂O oxidation pathway for H₂O₂ formation, rotating ring-disk electrode (RRDE) measurements are advised to determine the electron transfer number and H₂O₂ selectivity for the O₂ reduction pathway, thereby offering more comprehensive mechanistic insight.
- (9) The authors propose that H₂O₂ generation proceeds via indirect pathways involving radicals such as •OH or •O₂⁻. Scavenging experiments are recommended to further validate the contributions of these active species.
- (10) While the effect of pH on As³⁺ oxidation was examined, the influence of pH on the photocatalytic generation of H₂O₂ should also be investigated to fully assess the applicability of catalysts across varying reaction conditions.

Reviewer #2

(Remarks to the Author)

I have read the manuscript titled "Substituent coordination inducing oxidation-reduction molecular organic junction to boost heterogeneous interfacial hydrogen peroxide generation" by Li et al. with interest. The work addresses the construction of molecular organic frameworks with spatially separated redox centers to enhance photocatalytic H₂O₂ generation and arsenic removal. While the topic is relevant and potentially impactful, the manuscript suffers from conceptual, structural, and

analytical weaknesses that currently preclude publication in its present form. I recommend rejection at this stage, with the possibility of resubmission after major revision and significant additional experimental and analytical work.

My detailed comments are as follows:

The title is overly long and should be shortened for clarity and impact.

The abstract frames the problem unclearly and needs reorganization for logical flow.

In Fig. 1a, the schematic is confusing: the “reduction center” shows oxidation, and vice versa. The meaning of the arrows linking electrons and holes is ambiguous—do they indicate recombination?

The abstract mentions arsenic-containing wastewater purification, but this is not discussed elsewhere. The connection between catalytic performance and As adsorption/oxidation must be clarified.

In Fig. 2a, CTF-TF shows delocalized HOMO–LUMO orbitals. What specific role does fluorine substitution play in differentiating active sites?

The WOR is presented as endergonic; even for the N₂ site, the reaction remains non-spontaneous. How does it proceed beyond OH*? If via OH* dimerization, this step should be shown in the reaction coordinate diagram.

The claim that N₁ and N₂ are “confirmed” as oxidation and reduction centers is overstated. The evidence only suggests this assignment.

Optical data (Fig. S19) should be complemented with wavelength-dependent photocatalytic tests to determine the active excitation region.

The uniform distribution of active sites is asserted but not demonstrated. Structural or compositional evidence must be provided, as monomer substitution changes redox behavior.

The catalyst morphology (micrometer scale) raises concerns about diffusion and porosity effects. Please discuss how mass transport limits the rate and clarify what BET measurements truly probe.

The IR spectra (Fig. S14) and XPS fittings (Fig. S15) lack sufficient rigor. Clarify your peak identification and fitting criteria. The XPS fits, in particular, appear unreliable and must be redone.

H₂O₂ production data are not normalized by the number of active sites or surface area. Explain how variations in BET values are accounted for.

The non-linear dependence of reactivity on O₂ concentration (Fig. S56) needs explanation.

Radical quantification (Fig. 4f–g) shows large uncertainty, especially for OH•, and the proposed 2e⁻ ORR + 1e⁻ WOR mechanism appears charge-imbalanced. Please revisit this analysis.

The paragraph on As(III)/As(V) processes (lines 367–369) mixes redox and adsorption effects in a confusing way. These phenomena should be separated and discussed clearly.

The role of pH on in-situ H₂O₂ formation and As(III) oxidation kinetics should be addressed explicitly.

In summary, the manuscript presents an interesting idea but lacks coherence in its mechanistic rationale and supporting evidence. The experimental data are not sufficiently robust or consistent to substantiate the claims. Substantial reorganization, additional experiments, and clearer mechanistic validation would be required for the work to reach publishable quality.

Reviewer #3

(Remarks to the Author)

Reviewer #4

(Remarks to the Author)

The paper from Li et al. proposes a fluorine substituted molecular organic junction catalyst for efficient hydrogen peroxide photosynthesis. By tuning the coordination number of the F-substituted benzene linker, spatially separated oxidation-reduction centers were obtained to boost oxygen reduction and water oxidation reactions for hydrogen peroxide production. Overall, the paper is well written and proposes significant advancement to this field. Therefore, it is recommended for publication in Nature Communications after addressing the following minor issues.

Comments to authors

1. The authors should provide some explanation for the observed reduction in H₂O₂ production rates in real water compared to pure water.
2. The authors should add the important characterization data of CTF from Supplementary Information to main manuscript, since the innovation of this paper lies in catalyst design.
3. The authors should explain why the peak of FTIR will blue shift as pressure increases.
4. In-situ EPR is an important method to capture the continuous generation process of active species, and thus detailed test procedures and experimental photograph should be provided.
5. The authors should standardize the writing of all free radicals.

Version 1:

Reviewer comments:

Reviewer #1

(Remarks to the Author)

The revised manuscript addressed all the raised issues satisfactorily and I have no further comments.

Reviewer #2

(Remarks to the Author)

We have revised both the updated version of the manuscript as well as the provided rebuttal letter and, although we acknowledge that the quality of the manuscript has improved, there are still two crucial points to address.

1) In Figure 2B from the updated manuscript, authors present DFT calculations for ORR and WOR, in which the latter seems to be endergonic and thus, unfavoured for both active sites N1 and N2. Nonetheless, at figure S47, they show that for samples containing either full oxidation sites (CTF-1) or full reductive sites (CTF-TF), the H₂O₂ production does not seem to change dramatically, generating 0,2 and 0,15 mM for CTF-1 and CTF-TF, respectively. This has to be clarified experimentally with a probe reaction (e.g. silver reduction) if the manuscript is to be evaluated for publication.

2) When it comes to mass transport limitations, it is unclear whether reaction is occurring at the first layers (with plenty of material being inactive) or if the species diffuse in and out. The reason is because light could penetrate more than the reagents and the absorption event could occur at places where the reagents can not access, which is certainly important for photocatalytic applications. I understand the time constraints but the authors should include a sentence on the manuscript mentioning the likeliness of this effect.

Reviewer #3

(Remarks to the Author)

Reviewer #4

(Remarks to the Author)

The manuscript has been revised carefully according to comments and it is suitable for publication now.

Manuscript ID: NCOMMS-25-70315

Title: **Substituent-induced oxidation-reduction molecular organic junction for interfacial hydrogen peroxide photosynthesis**

Reviewer #1 (Remarks to the Author):

Summary Comments: The authors constructed a covalent triazine framework (CTF)-based catalysts featuring spatially separated oxidation and reduction centers. These catalysts exhibit exceptional performance in the photocatalytic synthesis of H₂O₂ and facilitate the direct oxidation of As³⁺ to As⁵⁺ using in-situ generated H₂O₂. However, several aspects require further attention:

Author response: We highly appreciate the positive comments from the reviewer, and they are all considered in corrected manuscript.

1. Although the introduction outlines the design concept of spatially separated redox centers and identifies the corresponding molecular structures, direct evidence confirming their respective roles as oxidation and reduction sites is lacking. Additional experimental or characterization data are necessary to substantiate this claim.

Author response: We appreciate the reviewer's critical comments. I sorry that the previous version did not mention the experimental evidence or characterization data regarding the oxidation and reduction sites in CTF-TF-0.5. Here, we have identified and reviewed a number of literatures on experimentally distinguishing oxidation and reduction sites. Currently, two primary experimental approaches are employed: **i, TEM analysis of oxidation or reduction sites.** Specifically, noble metals (platinum particles) are typically deposited as reduction products, while lead oxide or manganese oxide are deposited as oxidation products. After they were successfully deposited, TEM analysis was used to locate the position of these products to identify the respective sites where catalytic oxidation and reduction occur (*Appl. Catal. B: Environ.* 2021, 281, 119479; *Nat. Commun.* 2023, 14, 3901); **ii, Specific site poisoning experiments.** By introducing specific probe molecules to poison surface active sites, quantifying the products of corresponding oxidation and reduction reactions enables the assessment of the active role of oxidation and reduction sites (*Angew. Chem. Int. Ed.* 2021, 60, 16607-16614).

However, TEM analysis of oxidation or reduction sites is only applicable to lattice materials discernible under high-magnification TEM, such as materials with crystal faces exhibiting different redox activities or heterojunction materials containing distinct oxidation and reduction components. Specific site poisoning experiments are exclusively applicable to modified materials possessing specific active sites, such as active metal sites or organic functional groups.

Here, we conducted extensive literature research and considered using supplementary experiments or characterization data to demonstrate their respective roles as oxidation and reduction sites (the triazine ring connecting two benzene rings and one fluorinated benzene ring acts as a oxidation site, while the triazine ring connecting one benzene ring and two fluorinated benzene rings acts as an reduction

site) (**Figure 1c**). Unlike conventional oxidation and reduction sites, the redox sites in our proposed CTF-TF-0.5 are determined by the distinct chemical microenvironments surrounding the triazine ring. Therefore, both the oxidation and reduction sites in CTF-TF-0.5 are the triazine ring. Consequently, it is challenging to distinguish between the same structure using conventional methods such as TEM analysis of oxidation or reduction sites or specific site poisoning experiments. Existing techniques cannot effectively identify the active sites responsible for oxidation and reduction within the same structure with different redox properties. Therefore, in the revised version, we further confirm the proposed structure and separable redox sites in CTF-TF-0.5 through additional characterization and theoretical computational simulations, including solid-state NMR spectra, time-dependent density functional theory (TD-DFT) for electron-hole distribution and Density of states (DOS), combined with our previous HOMO-LUMO analysis (**Figure 2a**) and ΔG diagram (**Figure 2b**).

Figure 1c, Gradual increase in the number of F substitutions to tune the oxidation-reduction sites of CTFs.

Here, firstly, we demonstrate the chemical environment and structural integrity using solid-state NMR (^{13}C , ^1H , and ^{19}F) spectroscopy (**Figure 3a** and **Figure S17**). In the CP-MAS ^{13}C -NMR spectrum, the CTF-1 demonstrates the obvious carbon signals from the triazine units at 170.4 ppm (a) and the phenyl carbons (b and c) at 138.8 ppm and 128.7 ppm. In the CTF-TF, the carbon signal of triazine rings shifts to 159 ppm because of the introduction of fluorine atoms (a'), and the carbon signal of fluorinated benzene rings are located at 145.1 ppm (c'). Moreover, the carbon atoms connected with triazine rings display a peak at 119 ppm (b'). Note that the CTF-TF-0.5 simultaneously displayed the carbon signals (a and a') from the triazine units without and with fluorine atoms and the phenyl carbons (b and c), and the carbon signal (c') of fluorinated benzene rings, and the carbon atoms connected with triazine rings (b'), which indicated that CTF-TF-0.5 contains both a benzene ring and a fluorine-substituted benzene ring (*Nat. Commun.* 2023, 14, 8114).

The solid-state ^1H NMR spectra of CTF-1 and CTF-TF-0.5 exhibit a signal at approximately 7.5 ppm, originating from H atom on the benzene ring. The solid-state ^{19}F NMR spectra of CTF-TF-0.5 and CTF-TF exhibit a signal at approximately -144 ppm, originating from F atom on fluorine-substituted benzene ring. All these

characterizations fully indicate that CTF-1, CTF-TF-0.5 and CTF-TF was successfully synthesized.

Figure 3a Solid-state ^{13}C NMR spectra and possible structural model representation.

Figure S17: Solid-state ^1H (a) and ^{19}F (b) NMR spectra and possible structural model representation (c).

Time-dependent density functional theory (TD-DFT) calculations indicated that electrons are primarily localized in the triazine-connected region comprising one benzene and two F-substituted benzenes, while holes are predominantly localized in the triazine-connected region comprising two benzenes and one F-substituted benzene in CTF-TF-0.5 (**Figure S1**). In contrast, the electron and hole distribution regions on the surfaces of CTF-1 and CTF-TF exhibit no distinct characteristic patterns.

For catalytic reactions, the energy band positions of highest occupied molecular orbital (HOMO) and lowest unoccupied molecular orbital (LUMO) determine the oxidation and reduction capacities, and their spatial distribution determines the

distribution of oxidation-reduction site on the catalyst molecular. As shown in our manuscript, CTF-TF-0.5 exhibited a clear separated spatial distribution of HOMO and LUMO. Specially, its HOMO is localized at the triazine connected two benzene and a F-substituted benzene (N1), while LUMO is localized at the triazine connected a benzene and two F-substituted benzene (N2). In contrast, both CTF-1 and CTF-TF showed irregular and overlapping spatial distributions of LUMO and HOMO (**Figure 2a**).

Figure S1: Distribution of holes (blue) and electrons (green) in CTF-1, CTF-TF-0.5 and CTF-TF obtained through the time-dependent density functional theory (TD-DFT) calculations (Isosurface value = 0.001). The white, red, cyan, brown and gray colors represent hydrogen, oxygen, fluorine, carbon and nitrogen.

The DFT calculations using the HSE06 hybrid functional were also carried out for CTF-1, CTF-TF-0.5 and CTF-TF, and the band structures and projected density of states (PDOS) are shown in **Figure S2**. In CTF-TF-0.5, the VB and CB edges (HOMO and LUMO) have large DOS peaks, which are related to their molecular origin. The PDOS of the VB and CB edge simultaneously present more nitrogen orbital contributions. This suggests that the VB and CB originate from N on the triazine ring and trace amounts of C in CTF-TF-0.5. However, the elemental orbital distributions of the VB and CB edges (HOMO and LUMO) in CTF-1 and CTF-TF do not exhibit characteristic distributions.

It can be observed that the microenvironments of the triazine ring in CTF-TF-0.5

differ between N1 and N2. Furthermore, to accurately distinguish the contributions of N1 and N2 on the triazine ring to the VB and CB edges, we selected a single CTF-TF-0.5 single unit for DOS analysis. As shown in **Figure S3**, the PDOS of HOMO has higher N1 2p orbital contributions, while the LUMO has more N2 2p orbital contributions. This suggests that the VB comes from the triazine connected two benzene and a F-substituted benzene (N1), while the CB edge comes from the triazine connected a benzene and two F-substituted benzene (N2), which agrees with the molecular orbitals, as shown in **Figure 2a**, where the results of theoretical simulations of HOMO/LUMO and distribution of holes and electrons originate from the donor N2 and acceptor N1, respectively.

Figure S2 Density of states of CTF-1 (a), CTF-TF-0.5 (b) and CTF-TF (c). The inset show the DOS regions for LUMO and HOMO in CTF-TF-0.5.

Figure S3 Selected DOS analysis region for a single CTF-TF-0.5 unit (a). DOS of different N 2p with HOMO (b) and LUMO (c) region in single CTF-TF-0.5 unit.

To further verify the feasibility of H₂O₂ photosynthesis in CTFs with different oxidation and reduction sites, we investigated the thermodynamics of the WOR and ORR on CTF-1, CTF-TF-0.5 and CTF-TF by analyzing Gibbs free energy (ΔG) and intermediate configurations (**Figures 2b** and **S4-S11**). These results revealed that CTF-TF-0.5_N2 exhibited the lowest free energy for ORR to H₂O₂, accompanied by a sequential transition of reactive oxygen intermediates (*O₂⁻ to *OOH), followed by proton coupling to form H₂O₂. Meanwhile, CTF-TF-0.5_N1 showed the lowest free energy for WOR to H₂O₂, with a reactive oxygen intermediate (*OH), which then combines to produce H₂O₂. These confirmed that the N1 site of CTF-TF-0.5 serves as the oxidation site for WOR, while the N2 site acts as the reduction site for ORR. Moreover, CTF-TF-0.5 facilitates more efficient bulk charge separation of photogenerated carriers into distinct molecular units, enhancing H₂O₂ overall photosynthesis compared to CTF-1 and CTF-TF.

Based on the all above results, we proposed a spatially separable oxidation-reduction assignments in CTF-TF-0.5 that the N1 site of CTF-TF-0.5 serves as the oxidation site for WOR, while the N2 site acts as the reduction site for ORR.

Our modification to the manuscript: The corresponding description was added to the revised manuscript (**Pages 5 and 7, Figure 3a**) and Supplementary Information (**Figures S1-S3**). The specific calculation process has also been added in **Text S8**.

2. The reported H₂O₂ production activity was measured under simulated sunlight using a full-spectrum xenon lamp at 150 mW cm⁻² without an AM 1.5G filter. To accurately reflect solar conditions, performance should be evaluated under standard AM 1.5G irradiation, and the solar-to-chemical conversion (SCC) efficiency should be calculated to enable comparison with other catalytic systems.

Author response: We appreciate the reviewer's professional comments. Based your suggestion, we have added the data on the photocatalytic H₂O₂ production performance using the AM-1.5G filter with and without Us, and the solar-to-chemical conversion (SCC) efficiency with mechanical force was also calculated.

Figure S55 Time profiles of CTFs for H₂O₂ photosynthesis using the AM-1.5G filter with and without Us. Experimental conditions: catalyst (0.125 g L⁻¹) under Us and light, T = 25°C, water.

5 mg CTF-TF-0.5 was fully dispersed into the 100 mL beaker containing 40 mL aqueous solution in dark before photocatalytic reaction ultrasound. Then, the photocatalyst was stirred for 1 h in ambient condition under the irradiation by a xenon lamp (AM 1.5). The SCC efficiency (η) was calculated by [PNAS, 2021, 118, e2115666118]:

$$\eta(\%) = \frac{\Delta G_{H_2O_2} \times n_{H_2O_2}}{t_{ir} \times S_{ir} \times I_{AM}}$$

where $\Delta G_{H_2O_2}$ is the free energy for H_2O_2 production ($117 \times 10^3 \text{ J mol}^{-1}$), $n_{H_2O_2}$ is the amount of generated H_2O_2 (mol) and t_{ir} is the irradiation time (s). The overall irradiation intensity (I_{AM}) of the AM 1.5 global spectrum (300-2500 nm) is 0.1 W cm^{-2} and the irradiation area (S_{ir}) is $3.14 \times \text{cm}^2$.

The SCC of CTFs with and without mechanical force assistance was calculated in **Text S7** and **Table S4**. CTF-TF-0.5 with mechanical force exhibited the highest SCC of 0.16% relative to other scenarios, surpassing the typical photosynthetic efficiency of plants of 0.10% (*Nat. Commun.* 2025, 16, 7654), and this SCC demonstrated the excellent performance compared to the reported catalysts (**Table S5**).

Table S4. The comparison of the SCC efficiencies of CTF-1, CTF-TF-0.5 and CTF-TF.

Catalytic system	$n_{H_2O_2}$ (mol)	SCC (η , %)
CTF-TF-0.5/Us/Light	0.016	0.16
CTF-TF/Us/Light	0.0057	0.059
CTF-1/Us/Light	0.0075	0.077
CTF-TF-0.5/Light	0.0073	0.075
CTF-TF/Light	0.0030	0.031
CTF-1/Light	0.0025	0.026

Table S5. The comparison of the SCC efficiencies of CTF-TF-0.5 and other reported polymer photocatalysts.

Catalysts	SCC (%)	Ref.
g-C ₃ N ₄ /PDI	0.10	20
g-C ₃ N ₄ /BDI	0.13	21
g-C ₃ N ₄ /PDI/rGO	0.20	22
COF-TfpBpy	0.57 1.08	23
PM-CDs-30	0.21	24
Nv-C≡N-CN	0.23	25
APFac resins	0.54	78

TPT-alkynyl-AQ	0.35	26
rGO@MRF-0.5	1.23	79
SA-TCPP	1.2	80
CNIO-GaSA	0.4	27
RF-base resins	0.5	28
RF-acid-resins	0.7	81
RF/P ₃ HT	1	29
TPE-AQ	0.26	5
DE7-M	0.23	30
o-COF-TpPzda	0.46	46
PyIm-COF	0.28	34
TpPa/TpDz	1.34	35
TpPm	1.84	36
B[f]QCOF-1	0.23	37
COF-JLU90	1.52	38
TBC-COF	0.77	49
TTT-COF	0.32	41
TAPT-FTPB COFs	1.22	52
BTT-H3 COF	2.02	43
BTTA-COF	0.48	54
g-C ₃ N ₄	0.14	76
PHI	0.18	76
C ₅ N ₂	2.6	76
CTF-TF-0.5	0.16	This work

Our modification to the manuscript: The corresponding description was added to the revised manuscript (**Page 14**) and Supplementary Information (**Text S7, Figure S55 and Tables S4-S5**).

3. Given the light absorption of catalysts up to 600 nm, measuring the apparent quantum yield (AQY) at specific wavelengths across the 360-600 nm range is recommended. Comparative AQY values against established high-performance catalysts would strengthen the performance assessment.

Author response: We appreciate the reviewer's professional comments.

Apparent quantum yield measurements: For apparent quantum yield (AQY) measurements, 5 mg catalyst was dispersed in 40 mL. A 300 W Xe-lamp with a band-pass filter of 380±15 nm, 420±15 nm, 450±15 nm, 500±15 nm, 550±15 nm and 600±15 nm was used as the incident light source. The light intensity was adjusted to be 4.51 mW cm⁻², 3.21 mW cm⁻², 3.56 mW cm⁻², 5.93 mW cm⁻², 7.45 mW cm⁻² and 6.49 mW cm⁻², respectively. The irradiation area was controlled to be 3.14 cm². The amount of H₂O₂ production was analyzed after 1 h irradiation. AQY was calculated using the following equation:

$$\text{AQY}\% = 2 \times (N_{\text{H}_2\text{O}_2} \cdot N_A \cdot h \cdot c) / (I \cdot S \cdot t \cdot \lambda) \times 100\%$$

where $N_{\text{H}_2\text{O}_2}$ was the amount of H₂O₂ production (mol), N_A was the Avogadro constant ($6.022 \times 10^{23} \text{ mol}^{-1}$), h was the Planck constant ($6.626 \times 10^{-34} \text{ J}\cdot\text{s}$), c was the speed of light ($3 \times 10^8 \text{ m}\cdot\text{s}^{-1}$), I was the irradiation intensity ($\text{W}\cdot\text{cm}^{-2}$), S was the irradiation area (cm^2), t was the irradiation time (s) and λ was the wavelength of incident light (m).

The AQY of CTF-TF-0.5 was calculated at specific wavelengths and shown to approximately match with the UV-Vis spectrum (**Figure S51**). CTF-TF-0.5 exhibited an AQY of H₂O₂ production close to 23.9%, 24.1%, 17.6%, 6.9%, 4.3% and 3.4% at wavelengths of 380 nm, 420 nm, 450 nm, 500 nm, 550 nm and 600 nm, respectively, which demonstrated the excellent AQY at 420 nm compared to the reported catalysts (**Table S2**). Overall, CTF-TF-0.5 exhibited the high photocatalytic activity of H₂O₂ production.

Figure S51: Apparent quantum yield (AQY) of H₂O₂ production at specific wavelengths superimposed with its UV-Vis absorption curve.

Table S2. The comparison of AQY of CTF-TF-0.5 at 420 nm and other reported polymer photocatalysts.

Catalysts	AQY (%)	Ref.
-----------	---------	------

g-C ₃ N ₄ /PDI	2.6	20
g-C ₃ N ₄ /BDI	4.6	21
g-C ₃ N ₄ /PDI/rGO	6.1	22
COF-TfpBpy	8.1	23
PM-CDs-30	0.98	24
Nv-C≡N-CN	1.8	25
TPT-alkynyl-AQ	18 (425 nm)	26
CNIO-GaSA	7.1 (459 nm)	27
RF-base resins	7.8	28
RF/P ₃ HT	10.6	29
DE7-M	8.7	30
Por-HQ-COF	5.05	31
Py-OH-Sa COF	5.78	32
.ACOF-S-EtOH	13.3	33
PyIm-COF	3.7	34
TpPa/TpDz	55	35
TpPm	22.7	36
B[f]QCOF-1	8.9 (450 nm)	37
COF-JLU90	7.1	38
AlCCOF-1	7.1	39
COF-OMe	6.5 (400 nm)	40
TTT-COF	12 (400 nm)	41
DT ₂ TA-TAPB	2.8	42
BTT-H2 COF	18 (440 nm)	43
BTT-H3 COF	17.7 (467 nm)	43
COF-N32	6.2 (459 nm)	44
TTF@Por-COF-cya	14.98	45
CTF-TF-0.5	24.12	This work

Our modification to the manuscript: The corresponding description was added to the revised manuscript (**Page 13**) and Supplementary Information (**Text S6, Table S2 and Figure S51**).

4. While the catalyst demonstrates high H₂O₂ production activity, direct comparative data with existing systems are not provided. Inclusion of such comparisons would better contextualize its performance and highlight any competitive advantages.

Author response: We appreciate the reviewer's professional comments. Here, we provided a direct comparative data with existing systems. Since there are few reports on the production of H₂O₂ by applying both external force and light, I summarized the H₂O₂ production performance reported covalent organic frameworks photocatalysts, all piezoelectric catalysts and all piezoelectric photocatalysts. Therefore, we have further summarized the detailed reaction conditions including light intensity, ultrasound power and frequency, and rotational speed in all reference lists (**Table S3**) for further comparison. Overall, H₂O₂ yield we reported exceeds that of most catalysts.

Table S3. H₂O₂ production rates for CTF-TF-0.5 in this work compared with representative recently reported work.

Catal.	Condition	Light intensity	Ultrasound conditions	Sacrificial agent	H ₂ O ₂ (μmol g ⁻¹ h ⁻¹)	Ref.
Por-HQ-COF	Vis	-	-	-	1525	31
Py-OH-Sa COF	Light	-	-	10% IPA	4780	32
o-COF-TpPzda	Vis	-	-	O ₂	4396	46
COF-M180	sunlight	98 mW cm ⁻²	-	O ₂	12500	47
ACOF-S-EtOH	AM 1.5G	-	-	-	5440	33
PyIm-COF	Vis	-	-	-	5850	34
COF-IN-1	Vis	100 mW·cm ⁻²	-	-	5463	48
TpPa/TpDz	Vis	-	-	O ₂	24420	35
TpPm	Vis	-	-	-	17014	36
B[f]QCOFs	Vis	-	-	-	9025	37
COF-JLU90	AM 1.5 G	100 mW cm ⁻²	-	-	6432	38
AICCOF-1	Light	200 mW cm ⁻²	-	O ₂ 10% BA	16794.69	39
COF-OH	Vis	300 mW cm ⁻²	-	-	4458	40
TBD-COF	White LED	17 mW cm ⁻²	-	-	5448	49
COF/In ₂ S ₃	Light	-	-	-	5713.2	50
TTT-COF	AM 1.5	100 mW cm ⁻²	-	10% BA	29900	41

Hz-TP-BT-COF	Vis	-	-	-	5700	51
TAPT-FTPBCOFs	AM 1.5 G	-	-	-	3780	52
DT ₂ TA-TAPB	Vis	100 mW cm ⁻²	-	-	8587	42
PAFs-363	Vis	-	-	-	3930	53
BTT-H2	Vis	-	-	-	1588	43
COF-N32	Vis	100 mW cm ⁻²	-	-	605	44
BTTA-COF	Vis	200 mW cm ⁻²	-	-	2650	54
Por-COF- cya	Vis	-	-	-	2290	45
CNF/SCNF-MS	Us/Stirring	-	45KW	-	62.8	55
BTO NSs	Us	-	180 W, 35 kHz	10 vol% MeOH	125.59	56
Au/BiVO ₄	Us	-	120 W, 40 kHz	4-CP	344.4	57
BiOCl	Us	-	-	Tris-buffered solution	420	58
SiO ₂ /PVDF-HFP	Us	-	300 W, 40 kHz	20 vol% EtOH	492	59
BiOCl	Us/Stirring	-	150 W, 53 kHz	-	560	60
C ₃ N _{5-x} -O	Us	-	-	-	615	61
C ₃ N ₄	Us/Stirring	-	150 W, 53 kHz/ 300rpm	-	680	62
UBTO-OV2	Us	-	300 W, 40 kHz	10 vol% EtOH	1611.2	63
Bi ₁₂ O ₁₇ Cl ₂	Us	-	100 W, 40 kHz	-	7760	64
BON-M	Us	-	40 kHz, 240 W	-	970.27	65
MOC-AuNP/gC ₃ N ₄	Us	-	300 W	-	120.21	66
BF1.5	Us	-	40 kH, 200 W	-	1305	67
BC-10	Us	-	120 W, 40 kHz	-	1299	68
BIO	Us	-	152 W, 40 kHz	-	9200	69

LFZ	Us/Vis	-	180 W, 40 kHz	-	403	70
Bi ₄ NbO ₈ Br	Us/Vis	-	280 W, 40 kHz	10 vol% EtOH	792	71
Bulk-g-C ₃ N ₄	Us/Vis	-	240 W, 40 kHz	0.1 M glucose	1080	72
BaTiO ₃ :Nb/C	Us/Vis	100 mW cm ⁻²	150 W, 40 kHz	10 vol% EtOH	1360	73
COF-DH-Eth	Us/Vis	-	10W, 49KHz	-	9212	74
PCCN-10	Us/Vis	100 mW cm ⁻²	100 W, 40 kHz	10 vol% EtOH	4610	75
C ₅ N ₂	Us/Vis	100 mW cm ⁻²	400 W, 40 kHz	-	1343.6	76
UiO-66-AQ	Us/Light	-	200 W, 40 kHz	-	7872.4	77
CTF-TF-0.5	Us/ Simulating sunlight	150 mW cm ⁻²	100 W, 40 kHz	O ₂	4664	This work
CTF-TF-0.5	Us/AM-1.5G	100 mW cm ⁻²	100 W, 40 kHz	-	3114.4	This work

5. Although multiple characterization techniques were employed to verify the catalyst structure, solid-state NMR (¹³C, ¹H, and ¹⁹F) spectroscopy could provide further insight into the chemical environment and structural integrity.

Author response: We appreciate the reviewer's professional comments. In the revised version, we further demonstrate the chemical environment and structural integrity using solid-state NMR (¹³C, ¹H, and ¹⁹F) spectroscopy. The specific description has been displayed in our response to your Comment 1. See your Comment 1

Our modification to the manuscript: The corresponding description was added to the revised manuscript (**Page 7, Figure 3a**) and Supplementary Information (**Figure S17**).

6. The authors suggest that the catalyst can adsorb O₂ from air (Figure 4a). O₂-temperature-programmed desorption (O₂-TPD) experiments are recommended to quantitatively assess O₂ adsorption capacity and offer more direct evidence.

Author response: We appreciate the reviewer's professional comments. In the ORR, to produce H₂O₂, O₂ adsorption on the surface of the catalyst is the first step. The O₂ temperature-programmed desorption (O₂-TPD) curve reveals that CTF-TF-0.5 has a higher O₂ adsorption (0.65 m mol g⁻¹) than CTF-1 (0.20 m mol g⁻¹) and CTF-TF (0.14 m mol g⁻¹) (**Figure S61**).

Figure S61: O₂-TPD curves of CTF-TF-0.5, CTF-1 and CTF-TF.

Our modification to the manuscript: The corresponding description was added to Supplementary Information (**Figure S61**).

7. Figure 4c indicates a high H₂O₂ production rate over 6 hours, with an approximately linear increase and no sign of plateau. Extending the reaction duration would help determine the maximum H₂O₂ yield, while post-reaction characterization of the catalyst would aid in evaluating its stability.

Author response: We appreciate the reviewer's professional comments. CTF-TF-0.5 demonstrated excellent stability during the first six cycles. However, as the reaction progressed, the accumulation of H₂O₂ within the system gradually increased and stabilized at approximately 23.65 mmol g⁻¹·h⁻¹. The H₂O₂ generation rate decreased from 3.13 mmol g⁻¹·h⁻¹ during the initial hour to 0.72 mmol g⁻¹·h⁻¹ over the subsequent 10 h (**Figures 5d**). The FT-IR spectra showed little change before and after the reaction, further confirming its excellent stability for catalytic reactions (**Figures S57**).

Figure 5d: Stability of CTF-TF-0.5 over 10 h.

Figure S57: FT-IR spectrum of fresh and used CTF-TF-0.5.

Our modification to the manuscript: The corresponding description was added to the revised Manuscript (**Figure 5d**) and Supplementary Information (**Figures S57**).

8. Although isotope labeling experiments confirm the H₂O oxidation pathway for H₂O₂ formation, rotating ring-disk electrode (RRDE) measurements are advised to determine the electron transfer number and H₂O₂ selectivity for the O₂ reduction pathway, thereby offering more comprehensive mechanistic insight.

Author response: We appreciate the reviewer's professional comments. The electron transfer pathway was evaluated by rotating ring-disk electrode (RRDE) analysis at 1600 rpm in O₂ saturated phosphate buffer solution (0.1 M, pH 7) (*P. Natl. Acad. Sci.* 2021, 118, e2103964118). The ORR polarization curves are displayed in **Figure S63**. Compare with CTF-1 and CTF-TF, CTF-TF-0.5 shows the higher onset potential (0.38 V vs. 0.29 V and 0.23 V), and significant higher ring current (**Figure S63a**). A high H₂O₂ yield with 89% selectivity at 0.00 V (vs. RHE) is observed in CTF-TF-0.5 (**Figure S63b**). Besides, in contrast to CTF-1 and CTF-TF, CTF-TF-0.5 shows approximately two-electron (2e⁻) transfer (**Figure S63c**) from -0.2 V to 0.0 (vs. RHE). These results indicate triazole-based CTF-TF-0.5 has outstanding 2e⁻ ORR for H₂O₂ production with high selectivity.

Figure S63: The electrochemical production of H_2O_2 . **a**, ORR polarization curves. **b**, H_2O_2 selectivity. **c**, the corresponding number of transferred electrons.

Our modification to the manuscript: The corresponding description was added to the revised Manuscript (**Page 15**) and Supplementary Information (**Figures S63**).

9. The authors propose that H_2O_2 generation proceeds via indirect pathways involving radicals such as $\cdot\text{OH}$ or $\cdot\text{O}_2^-$. Scavenging experiments are recommended to further validate the contributions of these active species.

Author response: We appreciate the reviewer's professional comments. Based on your suggestion, 1, 2 and 5 mM of scavengers were added to the initial solution in photocatalytic H_2O_2 production with Us. TBA for $\cdot\text{OH}$ have little effect on the production of H_2O_2 . It is clearly visible, p-BQ for $\cdot\text{O}_2^-$ mainly contribute to H_2O_2 production, and with the increase of p-BQ, the production of H_2O_2 gradually decreases.

Figure S64: Scavenger tests of CTF-TF-0.5 in photocatalytic H_2O_2 production with Us at 60 min (TBA for $\cdot\text{OH}$, p-BQ for $\cdot\text{O}_2^-$, $C=1, 2, 5$ mM).

Our modification to the manuscript: The corresponding description was added to the revised Manuscript (Page 15) and Supplementary Information (Figure S64).

10. While the effect of pH on As^{3+} oxidation was examined, the influence of pH on the photocatalytic generation of H_2O_2 should also be investigated to fully assess the applicability of catalysts across varying reaction conditions.

Author response: We appreciate the reviewer's professional comments. H_2O_2 at millimolar level was produced over CTF-TF-0.5 over a wide pH range (pH 1–13), with higher yields obtained at lower pH (down to pH 3.0) due to elevated proton concentrations in acidic media (Figures S56). However, the H_2O_2 production activity significantly decreased when the pH was further lowered from pH=3 to pH=1, which may be due to that the occurrence of the competitive reaction of H_2 evolution ($2\text{H}^+ + 2\text{e}^- \rightarrow \text{H}_2$ (0 V vs. NHE)) along with the production of H_2O_2 (Nat. Commun. 2023, 14, 5742).

Figure S56: Time profiles of photocatalytic H_2O_2 production by CTF-TF-0.5 under Us over a wide pH range (pH=1, 3, 7, 10 and 13).

Our modification to the manuscript: The corresponding description was added to the revised Manuscript (Page 14) and Supplementary Information (Figure S56).

Thank you very much again for your kind and appropriate comments. We are sure that these comments help improve the quality of our manuscript significantly.

Sincerely yours,

Junhao Qin

Reviewer #2 (Remarks to the Author):

Summary Comments: I have read the manuscript titled “Substituent coordination inducing oxidation-reduction molecular organic junction to boost heterogeneous interfacial hydrogen peroxide generation” by Li et al. with interest. The work addresses the construction of molecular organic frameworks with spatially separated redox centers to enhance photocatalytic H₂O₂ generation and arsenic removal. While the topic is relevant and potentially impactful, the manuscript suffers from conceptual, structural, and analytical weaknesses that currently preclude publication in its present form. I recommend rejection at this stage, with the possibility of resubmission after major revision and significant additional experimental and analytical work. My detailed comments are as follows:

Author response: We appreciate the reviewer’s critical comments, which helped us to improve the overall quality of the manuscript significantly. By following the reviewer’s comments, we have revised the manuscript in a point-by-point manner. In particular, we tried to add more details and clarify ambiguous claims. Our responses and modifications are as follows:

1. The title is overly long and should be shortened for clarity and impact.

Author response: We appreciate the reviewer’s critical comments. The corresponding title has been revised using “Substituent-induced oxidation-reduction molecular organic junction for interfacial hydrogen peroxide photosynthesis”.

2. The abstract frames the problem unclearly and needs reorganization for logical flow.

Author response: We appreciate the reviewer’s professional comments. Based on your suggestion, the corresponding abstract has been revised as follows:

“The distribution of catalytic active sites critically dictates photocatalytic efficiency, but existing catalyst design operate at the same or adjacent sites still remain limitations toward photocatalytic reaction. To address this, a kind of spatially separable oxidation-reduction assignment in fluorine substituted molecular organic junction catalyst (covalent triazine framework, CTF-TF-0.5) is constructed.”

Our modification to the manuscript: The corresponding description was added to revised manuscript (**Page 1**).

3. In Fig. 1a, the schematic is confusing: the “reduction center” shows oxidation, and vice versa. The meaning of the arrows linking electrons and holes is ambiguous-do they indicate recombination?

Author response: We appreciate the reviewer’s professional comments. According to your suggestions, the corresponding Figures and expressions have been modified in the revised manuscript (**Figure 1a**).

Figure 1 | Concept of F substituent coordination inducing oxidation-reduction molecular organic junction to promote carrier separation. **a**, Separated oxidation-reduction centers promoting catalytic reaction with carrier separation at the dual active site. D and A: Donor and acceptor molecule. D⁺ and A⁻: Donor cation and acceptor anion. **b**, The proposed structures of oxidation-reduction sites. **c**, Gradual increase in the number of F substitutions to tune the oxidation-reduction sites of CTFs.

Our modification to the manuscript: The corresponding description was added to revised manuscript (**Page 4, Figure 1a**).

4. The abstract mentions arsenic-containing wastewater purification, but this is not discussed elsewhere. The connection between catalytic performance and As adsorption/oxidation must be clarified.

Author response: We appreciate the reviewer's professional comments. As you mentioned, I am very sorry that the previous version did not address the connection between catalytic performance and As adsorption/oxidation. Here, we further investigated the variation in in-situ H₂O₂ generation during all As oxidation processes. By incorporating experiments with different concentrations of As and quenching experiments during oxidation, we ultimately analyzed the relationship between As concentration for oxidation and in-situ H₂O₂ generation. Additionally, we probed the reaction mechanism of CTF-TF-0.5 for piezo-photocatalytic oxidation of As (III).

As shown in **Figures S75a** and **S75b**, CTF-TF-0.5 showed the highest oxidation rate of As (III) (77.6%) at 500 μg L⁻¹ with the rapid oxidation rate constant relative to that at 1000 μg L⁻¹ (58.8%) (**Figure S75c**), while the concentration of As (V) gradually increased. During the degradation, the production of H₂O₂ also shows a positive correlation with its oxidation capacity of As (III) (**Figure S75d**). The corresponding adsorption capacity was also shown in **Figure S76**.

Furthermore, we investigated the reaction mechanism of piezo-photocatalytic H₂O₂

production for As (III) oxidation. Scavengers were added to the initial solution in photocatalytic As (III) oxidation with Us. TBA for $\cdot\text{OH}$ have no effect on As (III) oxidation, while p-BQ for $\cdot\text{O}_2^-$ have some effect for that. Note that catalase for H_2O_2 mainly contribute to As (III) oxidation, due to that the addition of catalase resulted in its near-total loss of oxidation capacity (**Figure S77**). These results clearly demonstrate that As (III) oxidation via piezo-photocatalysis was directly utilized for the oxidation of As (III), without conversion into other reactive species.

Figure S75: a and b, Time profiles (a) and degradation rate (b) of CTF-TF-0.5 in 1 h for As (III) oxidation with different concentration. c, The first-order kinetics model of CTF-TF-0.5. d, in-situ generated H_2O_2 .

Figure S76: The oxidation amount of As (III) and the production and adsorption amounts of As (V) of CTF-TF-0.5 at 1 h with different concentration.

Figure S77: **a**, Time profiles (a) of CTF-TF-0.5 in 1 h for As (III) oxidation with different sacrificing agents. **b**, The corresponding histograms of As (III) and As (V) at 60 min. (2 mM TBA for $\cdot\text{OH}$, 2 mM p-BQ for $\cdot\text{O}_2^-$, 500 U/mL catalase for H_2O_2).

Based on all the above discussions, we find that as the initial concentration of As(III) increases, the catalytic performance of the in-situ generated H_2O_2 in the piezo-photocatalytic oxidation of As (III) gradually decreases. Furthermore, the catalytically generated H_2O_2 was directly utilized for As (III) oxidation without being converted into any intermediate active species.

Our modification to the manuscript: The corresponding description was added to revised manuscript (**Pages 18-19**) and Supplementary Information (**Figures S75-S77**).

5. In Fig. 2a, CTF-TF shows delocalized HOMO-LUMO orbitals. What specific role does fluorine substitution play in differentiating active sites?

Author response: We appreciate the reviewer's critical comments. The triazine ring (C_3N_3) is a nitrogen-rich aromatic heterocycle, endowing it with fundamental advantages as a material building block. It inherently functions as an electron-deficient aromatic ring. The highly electronegative nitrogen atoms on the ring induce electron withdrawal from the ring via inductive and conjugation effects, making it a natural electron reservoir (*Chem. Rev.* 2021, 121, 23, 14555-14593).

As shown in **Figure 1c**, unlike the bare CTF-1 without fluorine substitution, we obtained CTF-TF-0.5 and CTF-TF with varying degrees of fluorine substitution by controlling monomer substitution. As depicted in electrostatic potential in **Figure answering 1**, the F site exhibits a strongly electron-rich localized region compared to both the triazine and benzene rings. Consequently, the highly electronegative F atom induces electron density toward itself. At the molecular level, when fluorine substitutes the benzene ring, the attracted electrons are primarily "locked" around the C-F bond (*Nat. Commun.* 2025, 16, 11024; *Appl. Catal. B: Environ. Energy* 2026, 384, 126228). However, the triazine ring, acting as a bridge between the benzene ring and the fluorinated benzene ring, possesses a specific position and unique delocalized electron storage capacity. This allows the electrons locked at the C-F bond to rapidly encounter the entire π -conjugated system. Therefore, the number of triazine-linked fluorinated benzene rings significantly influences the molecule's capacity for electron storage and donation, thereby influencing the distribution of delocalized HOMO and LUMO orbitals.

Additional evidence was also used to support the role of F replacement, including solid-state NMR spectra, time-dependent density functional theory (TD-DFT) and Density of states (DOS), *etc.* See our response to your Comment 9

Figure 1c Gradual increase in the number of F substitutions to tune the oxidation-reduction center of CTFs.

Figure answering 1 Electrostatic potential distribution of CTF-1, CTF-TF-0.5 and CTF-TF

6. The WOR is presented as endergonic; even for the N_2 site, the reaction remains non-spontaneous. How does it proceed beyond OH^* ? If via OH^* dimerization, this step should be shown in the reaction coordinate diagram.

Author response: We appreciate the reviewer's professional comments. H_2O_2 is indeed formed through the dimerization of $*OH$. Here, combining our previous sacrificial agent experiments, gas shielding experiments, EPR, and in-situ EPR experiments (**Figure S62, S64, S66 and S71**), we have demonstrated the role of $*OH$ in WOR to synthesize H_2O_2 via piezo-photocatalysis.

According to your suggestion, the step of $*OH$ dimerization to form H_2O_2 was further simulated in the reaction coordinate diagram. Initially, after optimizing the adsorption sites of two $*OH$ on all CTFs (**Figures answering 2 and 3**), the most stable adsorption conformations were selected (Red), and ΔG diagram for the dimerization of two $*OH$ was calculated and added in the reaction coordinate diagram (**Figures 2b and S11**).

Figure answering 2 The top view and side view of 2 *OH in different adsorption conformations on CTF-1 (a) and CTF-TF (b) (Red selected as the optimal adsorption conformation).

Figure answering 3 The top view and side view of 2*OH in different adsorption conformations on CTF-TF-0.5_N1 (a) and CTF-TF-0.5_N2 (b) (Red selected as the optimal adsorption conformation).

Figure 2b: ΔG diagram of CTFs in the ORR and WOR reaction. The insets show the transition state in the ORR and WOR, and the white, red, cyan, brown and gray colors represent hydrogen, oxygen, fluorine, carbon and nitrogen.

Figure S11: Transition states of WOR for CTF-1, CTF-TF-0.5_N1, CTF-TF-0.5_N2 and CTF-TF.

Our modification to the manuscript: The corresponding description was added to revised manuscript (Page 5, Figure 2b) and Supplementary Information (Figure S11).

7. The claim that N_1 and N_2 are “confirmed” as oxidation and reduction centers is overstated. The evidence only suggests this assignment.

Author response: We appreciate the reviewer’s critical comments. Indeed, your suggestions are crucial for deeply understanding of our proposed oxidation and reduction centers.

In previous versions, we used delocalized HOMO-LUMO orbitals simulations and the reaction energy barriers occurring at the respective N sites of O_2 and H_2O to verify the assignment of oxidation and reduction centers. Here, we have also attempted to use experimental methods to substantiate our viewpoint by reviewing relevant reference. However, it is challenging to distinguish the redox sites-both of which are triazine rings-due to the differing chemical environments surrounding them, which affect their intrinsic activity. The specific analytical process is presented in our response for the

Comment 1 of Reviewer 1 and your Comment 9. Moreover, during this, we also added additional characterization and theoretical calculations to further substantiate the evidence supporting this assignment, including solid-state NMR spectra, time-dependent density functional theory (TD-DFT) and Density of states (DOS), etc.

To present our proposed viewpoint more objectively, we will rephrase this statement throughout the text using “oxidation and reduction assignment/site”.

Our modification to the manuscript: The corresponding description was modified to revised manuscript.

8. Optical data (Fig. S19) should be complemented with wavelength-dependent photocatalytic tests to determine the active excitation region.

Author response: We appreciate the reviewer’s professional comments.

Apparent quantum yield measurements: For apparent quantum yield (AQY) measurements, 5 mg catalyst was dispersed in 40 mL. A 300 W Xe-lamp with a band-pass filter of 380±15 nm, 420±15 nm, 450±15 nm, 500±15 nm, 550±15 nm and 600±15 nm was used as the incident light source. The light intensity was adjusted to be 4.51 mW cm⁻², 3.21 mW cm⁻², 3.56 mW cm⁻², 5.93 mW cm⁻², 7.45 mW cm⁻² and 6.49 mW cm⁻², respectively. The irradiation area was controlled to be 3.14 cm². The amount of H₂O₂ production was analyzed after 1 h irradiation. AQY was calculated using the following equation:

$$\text{AQY\%} = 2 \times (N_{\text{H}_2\text{O}_2} \cdot N_{\text{A}} \cdot h \cdot c) / (I \cdot S \cdot t \cdot \lambda) \times 100\%$$

where $N_{\text{H}_2\text{O}_2}$ was the amount of H₂O₂ production (mol), N_{A} was the Avogadro constant ($6.022 \times 10^{23} \text{ mol}^{-1}$), h was the Planck constant ($6.626 \times 10^{-34} \text{ J}\cdot\text{s}$), c was the speed of light ($3 \times 10^8 \text{ m}\cdot\text{s}^{-1}$), I was the irradiation intensity ($\text{W}\cdot\text{cm}^{-2}$), S was the irradiation area (cm^2), t was the irradiation time (s) and λ was the wavelength of incident light (m).

The AQY of CTF-TF-0.5 was calculated at specific wavelengths and shown to approximately match with the UV-Vis spectrum (**Figure S51**). CTF-TF-0.5 exhibited an AQY of H₂O₂ production close to 23.9%, 24.1%, 17.6%, 6.9%, 4.3% and 3.4% at wavelengths of 380 nm, 420 nm, 450 nm, 500 nm, 550 nm and 600 nm, respectively, which demonstrated the excellent AQY at 420 nm compared to the reported catalysts (**Table S2**). Overall, CTF-TF-0.5 exhibited the high photocatalytic activity of H₂O₂ production.

Figure S51: AQY of H₂O₂ production at specific wavelengths superimposed with its UV-Vis absorption curve.

Our modification to the manuscript: The corresponding description was added to the revised manuscript (**Page 13**) and Supplementary Information (**Text S6** and **Figure S51**).

9. The uniform distribution of active sites is asserted but not demonstrated. Structural or compositional evidence must be provided, as monomer substitution changes redox behavior.

Author response: We appreciate the reviewer's critical comments. I sorry that the previous version did not mention the experimental evidence or characterization data regarding the oxidation and reduction sites in CTF-TF-0.5. Here, we have identified and reviewed a number of literatures on experimentally distinguishing oxidation and reduction sites. Currently, two primary experimental approaches are employed: **i, TEM analysis of oxidation or reduction sites.** Specifically, noble metals (platinum particles) are typically deposited as reduction products, while lead oxide or manganese oxide are deposited as oxidation products. After they were successfully deposited, TEM analysis was used to locate the position of these products to identify the respective sites where catalytic oxidation and reduction occur (*Appl. Catal. B: Environ.* 2021, 281, 119479; *Nat. Commun.* 2023, 14, 3901); **ii, Specific site poisoning experiments.** By introducing specific probe molecules to poison surface active sites, quantifying the products of corresponding oxidation and reduction reactions enables the assessment of the active role of oxidation and reduction sites (*Angew. Chem. Int. Ed.* 2021, 60, 16607-16614).

However, TEM analysis of oxidation or reduction sites is only applicable to lattice materials discernible under high-magnification TEM, such as materials with crystal faces exhibiting different redox activities or heterojunction materials containing distinct oxidation and reduction components. Specific site poisoning experiments are exclusively applicable to modified materials possessing specific active sites, such as active metal sites or organic functional groups.

Here, we conducted extensive literature research and considered using

supplementary experiments or characterization data to demonstrate their respective roles as oxidation and reduction sites (the triazine ring connecting two benzene rings and one fluorinated benzene ring acts as a reduction center, while the triazine ring connecting one benzene ring and two fluorinated benzene rings acts as an oxidation center) (**Figure 1c**). Unlike conventional oxidation and reduction sites, the redox centers in our proposed CTF-TF-0.5 are determined by the distinct chemical microenvironments surrounding the triazine ring. Therefore, both the oxidation and reduction sites in CTF-TF-0.5 are the triazine ring. Consequently, it is challenging to distinguish between these sites using conventional methods such as TEM analysis of oxidation or reduction sites or specific site poisoning experiments. Existing techniques cannot effectively identify the active sites responsible for oxidation and reduction within the same structure with different redox properties. Therefore, in the revised version, we further confirm the proposed structure and separable redox sites in CTF-TF-0.5 through additional characterization and theoretical computational simulations, including solid-state NMR spectra, time-dependent density functional theory (TD-DFT) and Density of states (DOS), combined with our previous HOMO-LUMO analysis (**Figure 2a**) and ΔG diagram (**Figure 2b**).

Figure 1c, Gradual increase in the number of F substitutions to tune the oxidation-reduction sites of CTFs.

Here, firstly, we demonstrate the chemical environment and structural integrity using solid-state NMR (^{13}C , ^1H , and ^{19}F) spectroscopy (**Figure 3a** and **Figure S17**). In the CP-MAS ^{13}C -NMR spectrum, the CTF-1 demonstrates the obvious carbon signals from the triazine units at 170.4 ppm (a) and the phenyl carbons (b and c) at 138.8 ppm and 128.7 ppm. In the CTF-TF, the carbon signal of triazine rings shifts to 159 ppm because of the introduction of fluorine atoms (a'), and the carbon signal of fluorinated benzene rings are located at 145.1 ppm (c'). Moreover, the carbon atoms connected with triazine rings display a peak at 119 ppm (b'). Note that the CTF-TF-0.5 simultaneously displayed the carbon signals (a and a') from the triazine units without and with fluorine atoms and the phenyl carbons (b and c), and the carbon signal (c') of fluorinated benzene rings, and the carbon atoms connected with triazine rings (b'), which indicated that CTF-TF-0.5 contains both a benzene ring and a fluorine-substituted benzene ring (*Nat. Commun.* 2023, 14, 8114).

Figure 3a Solid-state ^{13}C NMR spectra and possible structural model representation.

Figure S17: Solid-state ^1H (a) and ^{19}F (b) NMR spectra and possible structural model representation (c).

The solid-state ^1H NMR spectra of CTF-1 and CTF-TF-0.5 exhibit a signal at approximately 7.5 ppm, originating from H atom on the benzene ring, while the solid-state ^{19}F NMR spectra of CTF-TF-0.5 and CTF-TF exhibit a signal at approximately -144 ppm, originating from F atom on fluorine-substituted benzene ring. All these characterizations fully indicate that CTF-1, CTF-TF-0.5 and CTF-TF was successfully synthesized.

Time-dependent density functional theory (TD-DFT) calculations indicated that electrons are primarily localized in the triazine-connected region comprising one

benzene and two F-substituted benzenes, while holes are predominantly localized in the triazine-connected region comprising two benzenes and one F-substituted benzene in CTF-TF-0.5 (**Figure S1**). In contrast, the electron and hole distribution regions on the surfaces of CTF-1 and CTF-TF exhibit no distinct characteristic patterns.

For catalytic reactions, the energy band positions of highest occupied molecular orbital (HOMO) and lowest unoccupied molecular orbital (LUMO) determine the oxidation and reduction capacities, and their spatial distribution determines the distribution of oxidation-reduction site on the catalyst molecular. As shown in our manuscript, CTF-TF-0.5 exhibited a clear separated spatial distribution of HOMO and LUMO. Specially, its HOMO is localized at the triazine connected two benzene and a F-substituted benzene (N1), while LUMO is localized at the triazine connected a benzene and two F-substituted benzene (N2). In contrast, both CTF-1 and CTF-TF showed irregular and overlapping spatial distributions of LUMO and HOMO (**Figure 2a**).

Figure S1: Distribution of holes (blue) and electrons (green) in CTF-1, CTF-TF-0.5 and CTF-TF obtained through the time-dependent density functional theory (TD-DFT) calculations (Isosurface value = 0.001). The white, red, cyan, brown and gray colors represent hydrogen, oxygen, fluorine, carbon and nitrogen.

The DFT calculations using the HSE06 hybrid functional were also carried out for CTF-1, CTF-TF-0.5 and CTF-TF, and the band structures and projected density of states (PDOS) are shown in **Figure S2**. In CTF-TF-0.5, the VB and CB edges (HOMO and LUMO) have large DOS peaks, which are related to their molecular origin. The PDOS of the VB and CB edge simultaneously present more nitrogen orbital contributions. This suggests that the VB and CB originate from N on the triazine ring and trace amounts of C in CTF-TF-0.5. However, the elemental orbital distributions of the VB and CB edges (HOMO and LUMO) in CTF-1 and CTF-TF do not exhibit characteristic distributions.

It can be observed that the microenvironments of the triazine ring in CTF-TF-0.5 differ between N1 and N2. Furthermore, to accurately distinguish the contributions of N1 and N2 on the triazine ring to the VB and CB edges, we selected a single CTF-TF-0.5 single unit for DOS analysis. As shown in **Figure S3**, the PDOS of HOMO has higher N1 2p orbital contributions, while the LUMO has more N2 2p orbital contributions. This suggests that the VB comes from the triazine connected two benzene and a F-substituted benzene (N1), while the CB edge comes from the triazine connected a benzene and two F-substituted benzene (N2), which agrees with the molecular orbitals, as shown in **Figure 2a**, where the results of theoretical simulations of HOMO/LUMO and distribution of holes and electrons originate from the donor N2 and acceptor N1, respectively.

Figure S2: Density of states of CTF-1 (a), CTF-TF-0.5 (b) and CTF-TF (c). The inset show the DOS regions for LUMO and HOMO in CTF-TF-0.5.

Figure S3: Selected DOS analysis region for a single CTF-TF-0.5 unit (a). DOS of different N 2p with HOMO (b) and LUMO (c) region in single CTF-TF-0.5 unit.

To further verify the feasibility of H₂O₂ photosynthesis in CTFs with different oxidation and reduction sites, we investigated the thermodynamics of the WOR and ORR on CTF-1, CTF-TF-0.5 and CTF-TF by analyzing Gibbs free energy (ΔG) and intermediate configurations (**Figures 2b** and **S4-S11**). The results revealed that CTF-TF-0.5_N2 exhibited the lowest free energy for ORR to H₂O₂, accompanied by a sequential transition of reactive oxygen intermediates (*O₂⁻ to *OOH), followed by proton coupling to form H₂O₂. Meanwhile, CTF-TF-0.5_N1 showed the lowest free energy for WOR to H₂O₂, with a reactive oxygen intermediate (*OH), which then combines to produce H₂O₂. These confirmed that the N1 site of CTF-TF-0.5 serves as the oxidation site for WOR, while the N2 site acts as the reduction site for ORR. Moreover, CTF-TF-0.5 facilitates more efficient bulk charge separation of photogenerated carriers into distinct molecular units, enhancing H₂O₂ overall photosynthesis compared to CTF-1 and CTF-TF.

Based on the all above results, we proposed a spatially separable oxidation-reduction assignments in CTF-TF-0.5 that the N1 site of CTF-TF-0.5 serves as the oxidation site for WOR, while the N2 site acts as the reduction site for ORR.

Our modification to the manuscript: The corresponding description was added to the revised manuscript (**Pages 5 and 7, Figure 3a**) and Supplementary Information (**Figures S1-S3**). The specific calculation process has also been added in **Text S8**.

10. The catalyst morphology (micrometer scale) raises concerns about diffusion and porosity effects. Please discuss how mass transport limits the rate and clarify what BET measurements truly probe.

Author response: We appreciate the reviewer's critical comments. Based on your suggestions, we have re-synthesized and tested the SEM (**Figure S13**) and specific surface area of CTF-TF-0.5. Consequently, we normalized the synthesis performance of H₂O₂ using specific surface area, demonstrating that catalytic performance does not stem from specific surface area or pore effects, but rather from the influence of the catalyst's molecular structure itself. Furthermore, your suggestion regarding the impact of mass transfer rates on catalyst performance is excellent; we will systematically present and analyze this in future work due to time constraints at present.

During the experiment, 5 mg catalyst was added in 40 mL pure water for subsequent experiments. Therefore, the experimental specific surface area of CTF-1, CTF-TF-0.5 and CTF-TF are 1.505, 2.130 and 1.925 m², respectively. Compared to CTF-1, the BET-characterized specific surface area showed a slight increase with the gradual increase in the number of F coordination, and the specific surface areas of CTF-1, CTF-TF-0.5, and CTF-TF were similar. Furthermore, after normalizing the H₂O₂ production performance based on specific surface area, CTF-TF-0.5 still exhibited the highest H₂O₂ production performance. The results of the BET test are ultimately intended to demonstrate that the specific surface area of different CTFs has little effect on H₂O₂ production performance.

Figure S13: SEM images of CTF-1 (a), CTF-TF-0.5 (b) and CTF-TF (c).

Figure S47: H₂O₂ production data normalized by surface area of all CTFs.

Our modification to the manuscript: The corresponding description was added to the revised manuscript (Page 13) and Supplementary Information (Figures S13 and S47).

11. The IR spectra (Fig. S14) and XPS fittings (Fig. S15) lack sufficient rigor. Clarify your peak identification and fitting criteria. The XPS fits, in particular, appear unreliable and must be redone.

Author response: We appreciate the reviewer's critical comments. Based on your suggestions, we have retested the FT-IR and re-simulated the IR spectrum and functional group vibrational regions using Gauss. Meanwhile, I also retest and refit the XPS.

i, IR spectra

The chemical bonding characteristics of CTF-1, CTF-TF-0.5, and CTF-TF were retested using Fourier transform infrared spectroscopy (FT-IR) (Figure 3b). All samples exhibited two characteristic vibration signals associated with triazine units, appearing in the range of 1300–1600 cm⁻¹. For CTF-1, sharp peaks at 1356 and 1515 cm⁻¹ corresponded to triazine unit vibrations, while the peak at 813 cm⁻¹ is attributed to benzene rings. Similarly, CTF-TF-0.5 and CTF-TF displayed peaks at 1356 and 1522 cm⁻¹, indicative of triazine units, along with two additional peak at 1484 and 983 cm⁻¹, assigned to the stretching vibrations of C–F bonds (*ACS Appl. Mater. Interf.* 2017, 9, 37731-37738), confirming the formation of CTF-TF and CTF-TF-0.5. Meanwhile, as the number of F

substituents increases, the peak at 812 nm gradually decreases, indicating that the benzene ring is progressively substituted by F as the amount of F substitution increases (From CTF-1 to CTF-TF-0.5, then to CTF-TF).

Then, we also re-simulated the IR spectrum and functional group vibrational regions using Gauss (**Figure S18** and **S19**). All simulated samples also exhibited two characteristic vibration signals associated with triazine units, appearing in the range of 1300–1600 cm^{-1} . For CTF-1, sharp peaks at 1395 and 1578 cm^{-1} corresponded to triazine unit vibrations. Similarly, CTF-TF-0.5 and CTF-TF displayed peaks at 1395 and 1578 cm^{-1} , indicative of triazine units, along with two additional peak at 1511 and 1022 cm^{-1} , assigned to the stretching vibrations of C–F bonds. Overall, the FT-IR spectra obtained from experimental data and those derived from computational simulations exhibit consistent trends, although some peaks show a slight blue shift relative to the FT-IR spectra obtained experimentally.

Figure 3b FT-IR spectra of all CTFs.

Figure S18: Simulated FT-IR spectra with using Gauss.

Figure S19: Molecular structure model constructed using Gaussian software. Blue represents N atoms, gray represents C atoms, white represents H atoms, and cyan represents F atoms.

Our modification to the manuscript: The corresponding description was added to the revised manuscript (**Page 7, Figure 3b**) and the Supplementary Information (**Figures S18-S19**).

ii, XPS fitting

Based on your suggestion, the corresponding XPS fitting has been redone and refitted. The survey X-ray photoelectron spectroscopy (XPS) spectra are shown in **Figure S20**, which displays that the prepared CTF is composed of C and N, and the prepared CTF-TF-0.5 and CTF-TF is composed of C, N and F. The small amount oxygen is due to the adsorbed CO₂ or H₂O on the material surface.

The C 1s XPS spectrum for CTF-1 (**Figure S21**) shows a sharp peak at 287.07 eV (assigned to the N-C=N group) and a sharp peak at 284.8 eV, assigning to the sp²-hybrid aromatic carbon in benzene ring. As the amount of F substitution gradually increases, the N-C=N bond in CTF-TF-0.5 and CTF-TF was reduced by approximately 1.1 eV, and the C-F gradually increases while the sp²-hybrid aromatic carbon in benzene ring progressively diminishes.

The N 1s XPS spectrum (**Figure S22**) of CTF-1 shows two peaks centered at 398.8 and 400.3 eV, corresponding to the N species in the triazine units and the N atoms from the triphenylamine moiety, respectively (*Angew. Chem.Int. Ed.* 2020, 59, 6007-6014; *ChemSusChem* 2019, 12, 4493-4499). As the F coordination number increases, the two peaks of N 1s exhibit the trend of increased binding energy, and the C-F peak significantly increase (**Figure S22** and **S23**).

Figure S20: Survey XPS spectra of CTF-1 (a), CTF-TF-0.5 (b) and CTF-T (c).

Figure S21: C 1s high-resolution XPS spectra of CTF-1 (a), CTF-TF-0.5 (b) and CTF-TF (c).

Figure S22: N 1s high-resolution XPS spectra of CTF-1 (a), CTF-TF-0.5 (b) and CTF-TF (c).

Figure S23: F 1s high-resolution XPS spectra of CTF-1 (a), CTF-TF-0.5 (b) and CTF-TF (c).

Our modification to the manuscript: The corresponding description was added to the Supplementary Information (**Figures S20-S23**).

12. H_2O_2 production data are not normalized by the number of active sites or surface area. Explain how variations in BET values are accounted for.

Author response: We appreciate the reviewer's critical comments. Based on your suggestions, the corresponding H₂O₂ production data have been normalized by surface area. These results demonstrate that CTF-TF-0.5 still exhibits the highest H₂O₂ synthesis performance under synergistic light and ultrasound conditions. During the experiment, 5 mg catalyst was added in 40 mL pure water for subsequent experiments. Therefore, the experimental specific surface area of CTF-1, CTF-TF-0.5 and CTF-TF are 1.505, 2.130 and 1.925 m², respectively. Compared to CTF-1, the BET-characterized specific surface area showed a slight increase with the gradual increase in the number of F coordination, and the specific surface areas of CTF-1, CTF-TF-0.5, and CTF-TF were similar. Furthermore, after normalizing the H₂O₂ production performance based on specific surface area, CTF-TF-0.5 still exhibited the highest H₂O₂ production performance. The results of the BET test demonstrated that catalytic performance does not stem from specific surface area or pore effects, but rather from the influence of the catalyst's molecular structure itself.

The specific description has been displayed in our response to your Comment 10. See your Comment 10

Our modification to the manuscript: The corresponding description was added to the Supplementary Information (**Figures S16 and S47**).

13. The non-linear dependence of reactivity on O₂ concentration (Fig. S56) needs explanation.

Author response: We appreciate the reviewer's critical comments. Nonlinear dependence can be interpreted as follows: We demonstrated that the mechanism of H₂O₂ synthesis primarily involves the formation of H₂O₂ through dual pathways of ORR and WOR, which was also confirmed by EPR and quenching experiments (**Figures S64-70**). Therefore, even H₂O₂ under N₂, WOR to synthesis H₂O₂ (H₂O → *OH → H₂O₂) still could occur, resulting in a small amount of H₂O₂ was also generated under N₂.

Figure S62: Effect of dissolved oxygen on H₂O₂ synthesis of CTF-TF-0.5 in 1 h.

14. Radical quantification (Fig. 4f–g) shows large uncertainty, especially for OH•, and the proposed $2e^-$ ORR + $1e^-$ WOR mechanism appears charge-imbalanced. Please revisit this analysis.

Author response: We appreciate the reviewer's critical comments.

i, Radical quantification (Fig. 4f – g) shows large uncertainty, especially for OH•

The instability and trace nature of signals are normal phenomena in detecting transient radicals using DMPO spin-trapping EPR techniques. Furthermore, the characteristic DMPO-•OH quadruplet (**Figures 5f–g**) observed in the study confirms •OH generation, which was also validated through EPR software simulation. Moreover, in our work, the water oxidation reaction accounted for a low proportion of H₂O₂ synthesis ($H_2O \rightarrow \bullet OH \rightarrow H_2O_2$), which was proved with sacrificial agent and atmosphere experiments (**Figure S62 and S64**). Therefore, the weak signal from •OH generation in WOR aligns with theoretical expectations and does not undermine the reliability of this detection result.

Figure S71: Simulated EPR spectrum ($\bullet O_2^-$, $\bullet OOH$ and $\bullet OH$) during H₂O₂ production with Us/light.

ii, the proposed $2e^-$ ORR + $1e^-$ WOR mechanism appears charge-imbalanced.

The proposed $2e^-$ ORR + $1e^-$ WOR mechanism refers to that the O₂ is localized at N2 reduces O₂ to undergo ORR through an indirect $2e^-$ transfer pathway with intermediates of $\bullet O_2^-$ and $\bullet OOH$ to form H₂O₂. Meanwhile, the positive holes localized at N1 oxidized H₂O to form $\bullet OH$ radicals in surface, which subsequently combined to produce H₂O₂ through a $1e^-$ WOR. This phenomenon of charge imbalance was attributed to competitive reactions occurring at the WOR site, such as the formation of $\bullet OH$ alongside WOR to produce O₂. Furthermore, we employed deuterated water as the reactant and subjected it to vacuum conditions to detect the reaction products to explore the reaction pathway for WOR.

Specially, CTF-TF-0.5 (1 mg) was added to H₂¹⁸O (2 mL) in a 20 mL glass tube which was sealed with a rubber stopper. After ultrasound for 1 h, the gaseous product was analyzed by gas chromatography-mass spectrometry (GC-MS). The liquid phase product was then extracted and the catalyst was removed with a filter tip. MnO₂ was then added to the liquid product and after 30 min, the gaseous product was analyzed

by GC-MS.

To investigate the H₂O₂ production via WOR pathway, an isotopic-labeling experiments were carried out using CTF-TF-0.5 as the photocatalyst and H₂¹⁸O as the water source. As shown in **Figure S67a**, after the piezo-photocatalysis, ¹⁸O₂ existed in the reaction system as gaseous products, suggesting that H₂¹⁸O could be oxidized to ¹⁸O₂ via a 4e⁻ oxidation pathway. Furthermore, H₂O₂ generated by piezo-photocatalysis was decomposed by MnO₂ produced the ¹⁸O₂ (**Figure S67b**), indicating that H₂¹⁸O was the raw material for piezo-photocatalytic H₂O₂ in the WOR pathway, suggesting that WOR could be also produce H₂O₂ via *OH dimerization with 1e⁻ WOR pathway, combining gas experiments (**Figure S62**) and quenching experiments (**Figure S64**). These indicate that the WOR simultaneously undergoes competing reactions: the oxidation of 4e⁻ to form O₂ and *OH dimerization with a 1e⁻ WOR to form H₂O₂.

Figure S67: a, GC-MS data of gaseous phase after piezo-photocatalysis of CTF-TF-0.5 with H₂¹⁸O (a) and gas products (b) after H₂O₂ decomposition with MnO₂.

Our modification to the manuscript: The corresponding references has been added in the revised manuscript (**Page 15**) and Supplementary Information (**Figures S67** and **S71**).

15. The paragraph on As(III)/As(V) processes (lines 367-369) mixes redox and adsorption effects in a confusing way. These phenomena should be separated and discussed clearly.

Author response: We appreciate the reviewer's critical comments. The redox and adsorption effects for As(III)/As(V) have been separated and discussed.

As shown in **Figures 6a** and **S72a**, CTF-TF-0.5 showed the highest oxidation rate of As (III) (63.5%) with the rapid oxidation rate constant relative to CTF-1 (26.8%) and CTF-TF (40.6%), while the concentration of As (V) of all CTFs gradually increased. During the degradation, the production of H₂O₂ also shows a positive correlation with its oxidation rate of As (III) (**Figures S72b** and **S72c**). Interestingly, the adsorption of As (V) in CTF-TF-0.5 was significantly higher than those of CTF-1 and CTF-TF, which was attributed to the excellent adsorption capacity of CTF-0.5 for As (V) (**Figure S72d**).

The effect of different pH on As oxidation was also evaluated in the **Figure 6c**, the concentration of As (III) was completely oxidized within 90 min at pH=3 and displayed a highest oxidation rate of 0.0527 min⁻¹ (**Figure S74a**), while As(V) could be adsorbed up to 54.2% at 3h, which were both significantly higher than those at pH=7 and pH=10.

These indicated that the oxidation of As (III) increased significantly and As (V) was adsorbed in large quantities with the decrease of pH. During 3 h reaction, it also exhibits a continuously increasing H₂O₂ concentration at pH=3, reaching 471.5 μM at 3 h, which was significantly higher than that at pH=7 and pH=10 (**Figures S74b and S74c**). The adsorption of As (V) in CTF-TF-0.5 at pH=3 was significantly higher than those at pH=7 and pH=10, which was due to the adsorption of As (V) could be promoted under acid condition (**Figure S74d**).

Figure 6 | Evaluation of environmental applications for mine wastewater treatment. **a** and **b**, Time profiles of all CTFs in 1 h (**a**) and CTF-TF (**b**) in 3 h for heterogeneous interfacial As (III) oxidation. **c**, The effect of different pH. **d**, As-containing mine wastewater treatment. The inset shows the photographs of wastewater during the degradation process.

Figure S72d: The oxidation amount of As (III) and the production and adsorption amounts of As (V) of CTF-1, CTF-TF-0.5 and CTF-TF at 1 h.

Figure S74d: The oxidation amount of As (III) and the production and adsorption amounts of As (V) of CTF-TF-0.5 at pH=3, 7 and 10 over 3 h.

Our modification to the manuscript: The corresponding description was added in the revised manuscript (**Pages 18-19**) to Supplementary Information (**Figures S72d and S74d**).

16. The role of pH on in-situ H₂O₂ formation and As (III) oxidation kinetics should be addressed explicitly.

Author response: We appreciate the reviewer's professional comments.

Based on your suggestion, we investigated the performance of in-situ H₂O₂ generation and As(III) oxidation during reaction at different pH. As shown in **Figure 6c** and **Figure S74a**, the concentration of As (III) was completely oxidized within 90 min at pH=3 and displayed a highest oxidation rate of 0.0527 min⁻¹ than those at pH=7 and pH=10. During 3 h reaction, it also exhibits a continuously increasing H₂O₂ concentration at pH=3, reaching 471.5 µM at 3 h, which was significantly higher than that at pH=7 and pH=10 (**Figures S74b and S74c**).

These results indicated that low pH significantly enhanced in-situ H₂O₂ production, as the protonated environment promotes ORR to generate H₂O₂ (O₂ + 2H⁺ → H₂O₂), which was also seen in our response for the Comment 10 of Reviewer 1. (*Nat. Commun.* 2023, 14, 5742). Concurrently, analysis of active species during the process of As (III) oxidation reveals that H₂O₂ serves as the primary active species to oxidize As (III), without conversion into other reactive species (The specific description has been displayed in our response to your Comment 4. See your Comment 4). Therefore, lowering the pH efficiently produced H₂O₂ through piezo-photocatalysis, thereby enhancing the oxidation capacity of As(III) and accelerating the oxidation kinetics of As (III).

Figure 6 | Evaluation of environmental applications for mine wastewater treatment. **a** and **b**, Time profiles of all CTFs in 1 h (**a**) and CTF-TF (**b**) in 3 h for heterogeneous interfacial As (III) oxidation. **c**, The effect of different pH. **d**, As-containing mine wastewater treatment. The inset shows the photographs of wastewater during the degradation process.

Figure S74: CTF-TF-0.5 for piezo-photocatalytic As (III) oxidation at pH=3, 7 and 10. The first-order kinetics model of As (III) oxidation (**a**). In-situ generated H_2O_2 (**b**) and the corresponding zero-order kinetics model (**c**). The oxidation amount of As (III) and the production and adsorption amounts of As (V) at 1 h (**d**).

Our modification to the manuscript: The corresponding description was added in the revised manuscript (**Pages 18-19**) and Supplementary Information (**Figure S74**).

Summary Comments: In summary, the manuscript presents an interesting idea but lacks coherence in its mechanistic rationale and supporting evidence. The experimental data are not sufficiently robust or consistent to substantiate the claims. Substantial reorganization, additional experiments, and clearer mechanistic validation would be required for the work to reach publishable quality.

Author response: We highly appreciate the comments from the reviewer, and they are all considered in corrected manuscript. In the revised version, we further demonstrate our structure and claims using Solid state NMR, Time-dependent density functional theory (TD-DFT) calculations, DOS calculations, GC-MS and additional extensive performance testing.

Thank you very much again for your kind and appropriate comments. We are sure that these comments help improve the quality of our manuscript significantly.

Sincerely yours,

Junhao Qin

Reviewer #3 (Remarks to the Author):

Summary Comments: I co-reviewed this manuscript with one of the reviewers who provided the listed reports. This is part of the Nature Communications initiative to facilitate training in peer review and to provide appropriate recognition for Early Career Researchers who co-review manuscripts.

Author response: We appreciate the reviewer's critical comments, which helped us to improve the overall quality of the manuscript significantly. By following the reviewer's comments, we have revised the manuscript in a point-by-point manner. In particular, we tried to add more details and clarify ambiguous claims in Responses to Reviewer #2.

Thank you very much again for your kind and appropriate comments. We are sure that these comments help improve the quality of our manuscript significantly.

Sincerely yours,

Junhao Qin

Reviewer #4 (Remarks to the Author):

Summary Comments: The paper from Li et al. proposes a fluorine substituted molecular organic junction catalyst for efficient hydrogen peroxide photosynthesis. By tuning the coordination number of the F-substituted benzene linker, spatially separated oxidation-reduction centers were obtained to boost oxygen reduction and water oxidation reactions for hydrogen peroxide production. Overall, the paper is well written and proposes significant advancement to this field. Therefore, it is recommended for publication in Nature Communications after addressing the following minor issues.

Author response: We highly appreciate the positive comments from the reviewer, and they are all considered in corrected manuscript.

1. The authors should provide some explanation for the observed reduction in H₂O₂ production rates in real water compared to pure water.

Author response: Thank you for the insightful suggestion. The reason of the lower rate of H₂O₂ production could be attributed to due to the presence of numerous ions or organic pollutant in real water. The corresponding statements are added in the revised manuscript as follows: "As shown in **Figure 5e**, the performance of CTF-TF-0.5 in the photocatalytic production of H₂O₂ can still be maintained at a high level under different solute conditions, despite slightly lower performance than pure water due to the impact of a large number of ions or organics in the real water on H₂O₂ production."

Our modification to the manuscript: The corresponding description has been added in the revised manuscript (**Page 14**).

2. The authors should add the important characterization data of CTF from Supplementary Information to main manuscript, since the innovation of this paper lies in catalyst design.

Author response: We thank you for the reviewer's constructive comment. The corresponding important characterization data has been added to the revised manuscript from Supplementary Information.

Our modification to the manuscript: The corresponding description has been added in the revised manuscript, including Solid state NMR, FT-IR and Solid state EPR (**Page 8, Figure 3**).

3. The authors should explain why the peak of FTIR will blue shift as pressure increases.

Author response: We appreciate the reviewer's critical comments. Thank you very much for your professional suggestion. In the case of organic polymers, pressurization shortens the distance between molecules and increases intermolecular repulsive forces and interactions, resulting in a blue shift in the vibrational frequency of FT-IR [*J. Mol. Struct.* 2008, 889, 1-11; *J. Mol. Liq.* 2002, 101, 149-158; *J. Chem. Phys.* 2002, 116, 2928-2935]. Therefore, owing to the spatial separated redox site of in CTF-TF-0.5, pressurization increases its molecular repulsion even more, resulting in a greater degree of blue-shift of CTF-TF-0.5 relative to CTF and CTF-TF.

4. *In-situ* EPR is an important method to capture the continuous generation process of active species, and thus detailed test procedures and experimental photograph should be provided.

Author response: We appreciate the reviewer's professional comments. We added the corresponding description in **Figure S68** and **Text S4**. Electron paramagnetic resonance (EPR, Bruker Corporation, EMX nano) was used to explore the free radicals using at ambient temperature. 100 μL DMPO was added to 5 mL reaction solution (Dimethyl sulfoxide (DMSO) was used as a solution to $^*\text{O}_2^-$, while water is used as a solution to capture $^*\text{OH}$). Ultrasonic force and light were applied to the sample via an ultrasonic cleaner and a 300 W Xe lamp. During the reaction, solution from the reactor was transferred at 20 rpm via a peristaltic pump to the EPR sample chamber for collecting signals, then returned to the reactor. All the samples were measured under the same conditions (Mod Amp: 1.000 G, Mod Freq: 100 kHz, Res Center: 125 mm, Res Length: 25 mm, Sweep time per sample: 30 s for offline samples and 2.5 s for samples in situ measurement ; Sample interval: 0.5 The spectra were fitting by EPR software.

Figure S68: Photograph of in-situ EPR system for photocatalytic H_2O_2 production with Us.

Our modification to the manuscript: The corresponding description has been added in the revised Supplementary Information (**Figure S68** and **Text S4**).

5. The authors should standardize the writing of all free radicals.

Author response: We appreciate the reviewer's professional comments. Thank you very much for your professional suggestion. According to your suggestion, all free radical writing is uniformed.

Our modification to the manuscript: The corresponding description was modified to

the revised manuscript and Supplementary Information.

Thank you very much again for your kind and appropriate comments. We are sure that these comments help improve the quality of our manuscript significantly.

Sincerely yours,

Junhao Qin

Manuscript ID: NCOMMS-25-70315A

Title: Substituent-induced oxidation-reduction molecular organic junction for interfacial hydrogen peroxide photosynthesis

Reviewer #1 (Remarks to the Author):

Summary Comments: The revised manuscript addressed all the raised issues satisfactorily and I have no further comments.

Author response: We appreciate for your recommendation of acceptance of this manuscript.

Thank you very much again for your kind and appropriate comments.

Sincerely yours,

Junhao Qin

Reviewer #2 (Remarks to the Author):

Summary Comments: We have revised both the updated version of the manuscript as well as the provided rebuttal letter and, although we acknowledge that the quality of the manuscript has improved, there are still two crucial points to address.

Author response: We appreciate the reviewer's critical comments, which helped us to improve the overall quality of the manuscript significantly. By following the reviewer's comments, we have revised the manuscript in a point-by-point manner. In particular, we tried to add more details and clarify ambiguous claims. Our responses and modifications are as follows:

1. In Figure 2B from the updated manuscript, authors present DFT calculations for ORR and WOR, in which the latter seems to be endergonic and thus, unfavoured for both active sites N1 and N2. Nonetheless, at figure S47, they show that for samples containing either full oxidation sites (CTF-1) or full reductive sites (CTF-TF), the H₂O₂ production does not seem to change dramatically, generating 0,2 and 0,15 mM for CTF-1 and CTF-TF, respectively. This has to be clarified experimentally with a probe reaction (e.g. silver reduction) if the manuscript is to be evaluated for publication.

Author response: We appreciate the reviewer's critical comments. We fully understand your request to experimentally verify our claim regarding redox sites in CTF-TF-0.5 using a probe reaction (e.g., silver reduction). Here, we must first clarify that neither CTF-1 nor CTF-TF exhibited discernible redox site distributions in any of our previous versions, and neither our textual descriptions nor experimental data substantiated this claim. We have only proposed that CTF-TF-0.5 exhibits separable redox site distribution: the oxidation site (triazine connecting two benzene rings and one F-substituted benzene) and the reduction site (triazine connecting one benzene ring and two F-substituted benzene rings) (Figure 1c). Therefore, it is inappropriate to state that CTF-1 contains full oxidation sites (CTF-1) and CTF-TF contains full reductive sites. Note that the separable oxidation-reduction sites exists only in CTF-TF-0.5, as previously demonstrated through Figures 2 and S1-S3. Consequently, conducting probe experiments (e.g., silver reduction) to investigate their impact for all CTFs on H₂O₂ production is not particularly meaningful.

Furthermore, we previously stated that existing techniques cannot effectively identify the active sites responsible for oxidation and reduction within the same structure with different redox properties in CTF-TF-0.5. The specific description has been displayed in our previous response to your Comment 9. We have confirmed the separable redox sites in CTF-TF-0.5 through additional characterization and theoretical computational simulations, including solid-state NMR spectra, time-dependent density functional theory (TD-DFT) and Density of states (DOS), combined with our previous HOMO-LUMO analysis and ΔG diagram. We hope that in the future, with the development of the chemical discipline, more advanced in-situ experimental methods can be developed to identify such active sites.

2. When it comes to mass transport limitations, it is unclear whether reaction is occurring at the first layers (with plenty of material being inactive) or if the species diffuse in and out. The reason is because light could penetrate more than the reagents

and the absorption event could occur at places where the reagents can not access, which is certainly important for photocatalytic applications. I understand the time constraints but the authors should include a sentence on the manuscript mentioning the likeliness of this effect.

Author response: We appreciate the reviewer's professional comments. Based on your suggestion, the corresponding description has been revised as follows:

"In addition, the mass transfer of reactants to the surface of photocatalyst is also an important factor affecting the reaction efficiency, which will be studied in our subsequent research."

Our modification to the manuscript: The corresponding description was added to revised manuscript (**Page 10**).

Thank you very much again for your kind and appropriate comments. We are sure that these comments help improve the quality of our manuscript significantly.

Sincerely yours,

Junhao Qin

Reviewer #3 (Remarks to the Author):

Summary Comments: I co-reviewed this manuscript with one of the reviewers who provided the listed reports. This is part of the Nature Communications initiative to facilitate training in peer review and to provide appropriate recognition for Early Career Researchers who co-review manuscripts.

Author response: We appreciate the reviewer's critical comments, which helped us to improve the overall quality of the manuscript significantly. By following the reviewer's comments, we have revised the manuscript in a point-by-point manner. In particular, we tried to add more details and clarify ambiguous claims in Responses to Reviewer #2.

Thank you very much again for your kind and appropriate comments. We are sure that these comments help improve the quality of our manuscript significantly.

Sincerely yours,

Junhao Qin

Reviewer #4 (Remarks to the Author):

Summary Comments: The manuscript has been revised carefully according to comments and it is suitable for publication now.

Author response: Thank you very much again for your kind and appropriate comments.

Sincerely yours,

Junhao Qin